# Menstrual changes after COVID-19 vaccination among menstruators of reproductive age: A cross- sectional study from Erbil City, Iraq

**Warda Hassan Abdullah** *

Department of Midwifery, College of Nursing, Hawler Medical University, Erbil, Kurdistan Region, Iraq

* warda.abdullah@hmu.edu.krd

**Data Availability Statement:** All relevant data are within the paper and Supporting Information files.

**Funding:** The author received no specific funding for this work.

## Abstract

The COVID-19 vaccination has been reported to have various post-vaccination effects, including potential changes in the menstrual cycle among menstruators of reproductive age. The aim of this study is to investigate the relationship between COVID-19 vaccination and menstrual changes among menstruators within Iraqi society and contribute to the knowledge about COVID-19 vaccine-related menstrual changes. The cross-sectional study involved a sample of 400 menstruators residing in Erbil City, Kurdistan Region, Iraq. The study specifically focused on individuals who were both infected and uninfected by COVID-19 and had received the COVID-19 vaccine. Individuals with hormonal diseases or those who had undergone hormonal therapy were excluded from the study. To ensure representation from different areas of Erbil City, the study was conducted in four primary healthcare centers selected based on geographic direction: South, East, North, and West sides. The author utilized a structured questionnaire to assess and compare changes in the menstrual cycle before and after receiving the COVID-19 vaccine. The study found that a significant proportion (71.8%) of the participants experienced menstrual changes after receiving the COVID-19 vaccination, particularly after receiving both doses. These changes included alterations in the characteristics and symptoms of the menstrual cycle compared to before vaccination. The findings suggest that the COVID-19 vaccination can potentially affect the menstrual cycle in women of reproductive age. In conclusion, by revealing distinct changes in cycle characteristics and symptomatology, the study findings contribute to an expanding body of evidence supporting the connection between vaccination and menstrual alterations. Future studies with larger sample sizes and diverse populations are warranted to validate and expand upon the results of this study.

## Introduction

The COVID-19 pandemic has presented an unprecedented global health crisis, affecting communities worldwide. In response, countries have implemented vaccination campaigns to

**Competing interests:** The authors have declared that no competing interests exist.

mitigate the spread of the virus and protect public health [1,2]. Like many other nations, Iraq has been carrying out COVID-19 vaccination efforts to protect the population from the detrimental effects of the virus. However, as with any novel medical intervention, it is crucial to comprehensively evaluate its potential impact on various aspects of health, including those specific to different demographic groups [3].

Historically, data collection and medical research have overlooked women's health, leading to the frequent misdiagnosis or absence of diagnosis for many women [4]. Conditions such as autism and heart attacks have primarily been understood from a male perspective, leading to a lack of recognition and understanding of the unique manifestations that women may experience [5]. Unfortunately, this neglect of women's health concerns extends to the testing and evaluation of the COVID-19 vaccine, perpetuating the invisibility of menstrual health as a crucial aspect of women's well-being [6].

Menstruation is a fundamental physiological process unique to individuals of reproductive age, and any changes in menstrual patterns can significantly impact their overall well-being [7]. Recent studies conducted in various countries have suggested potential associations between COVID-19 vaccination and menstrual changes. For instance, several studies have reported an increased prevalence of menstrual irregularities and changes in menstrual patterns, such as alterations in cycle length and flow volume, following COVID-19 vaccination [8–13]. However, these studies were conducted in different settings and populations. The impact of vaccination on menstrual health within the specific context of Iraq remains largely unexplored [14].

Moreover, it is important to consider the perspective and needs of the Iraqi society when conducting such studies. In Iraq, menstruation remains a topic that is often surrounded by cultural taboos and misconceptions. Menstruators face unique challenges and barriers when seeking healthcare services related to menstrual health [15]. Therefore, conducting an inquiry into post-COVID-19 vaccination menstrual alterations in Erbil City, Iraq, is significant as it addresses the specific needs and concerns of the Iraqi populace. The aim of this study is to identify the relationship between COVID-19 vaccination and menstrual changes among menstruators.

## Method

### Design and settings, of the study

A cross-sectional study was conducted at four primary healthcare centers situated in different geographic directions: South, East, North, and West sides. Erbil City, the capital of the Kurdistan Region in Iraq, served as the setting for the study. Primary healthcare centers were chosen as the research sites due to the fact that individuals of reproductive age typically seek health services at these facilities.

### Participants of the study

A convenience sample of 400 menstruators, ranging from 18 to 49 years old, who had a regular menstrual cycle and were both infected and uninfected by COVID-19, and vaccinated with the COVID-19 vaccine no longer than 3 months prior, were included in this study. The inclusion of participants who received the COVID-19 vaccine within the past 3 months allows for an examination of potential menstrual changes that may have occurred as a result of the vaccination. This timeframe enables a focused analysis of the short-term effects of the COVID-19 vaccine on menstruation.

By including participants vaccinated within the past 3 months, the study aims to capture any immediate menstrual changes that could be associated with the vaccination. This selection

ensures that any observed menstrual changes are more likely to be attributed to the vaccine, as the effects on the menstrual cycle may be more prominent in the immediate post-vaccination period.

Participants with major psychiatric disorders, hormonal diseases, or those receiving hormonal therapy, as well as anticoagulant therapy during and after their infection, were excluded from the study to ensure the clarity and specificity of the results.

In this study, used the term "menstruator" to refer to individuals who experience menstruation. This term is intentionally chosen to be inclusive of all individuals who may menstruate, regardless of their gender identity. Recognizing that not all individuals who menstruate identify as women, the term "menstruator" avoids excluding those who may identify as non-binary, transgender men, or individuals of other gender identities.

## Estimation of sample size

The sample size for this study was calculated using the SSC program (Sample Size Calculator) version 3.0.43. The following parameters were entered into the program: a confidence level (Cl) of 95%, a margin of error of 0.05%, and a population proportion of 50%. A proportion of 50% is often considered a conservative estimate in situations where there is limited prior knowledge or information about the variable being studied. Based on these inputs, the SSC program estimated a sample size of 385 menstruators. To ensure reliability and account for any potential dropouts or incomplete data, a slightly larger sample size of 400 menstruators was included in the study. This larger sample size provides an extra buffer and increases the robustness of the study's findings.

## Method and tool data collection

An interview questionnaire was designed for the purpose of data collection from menstruators. The questionnaire was developed after an extensive review of current local and international literature, including textbooks, articles, and scientific journals. This literature review helped the researcher gain a comprehensive understanding of the problem and guided the process of tool design. To ensure the content validity of the questionnaire, the researcher sought expert validation by presenting the questionnaire to a panel of 10 experts from different relevant specialties within the Kurdistan Region. This process involved preliminary consultations, questionnaire drafting, expert feedback integration. These experts included professionals in fields such as Maternity Nursing, Community Health Nursing, Obstetrics, Infectious Diseases, and Maternity Nurse Practitioners and experts were chosen based on their academic accomplishments and professional experience, ensuring diverse perspectives. The panel of experts was asked to review the questionnaire for its content clarity, relevance of items, and overall adequacy.

The experts' responses were evaluated based on their agreements or disagreements regarding the relevancy of items, clarity of content, and adequacy of the entire questionnaire. Out of the 10 experts, 10 agreed upon the items of the questionnaire with only a few suggested changes. The researcher took their responses and suggestions into consideration and made the necessary modifications to the tool. As a result, the questionnaire will consist of two parts. Part I will focus on demographic data, including personal characteristics such as age, level of education, and residential status. Part II will gather information related to being infected by Covid-19, Covid-19 vaccination, and menstrual cycle.

By engaging experts in the field and incorporating their feedback, the questionnaire was refined to ensure its content validity and relevance to the research topic. This meticulous

process enhances the reliability and validity of the data collected through the questionnaire, contributing to the overall robustness of the study.

The study was conducted at Primary Health Care Centers, where the researcher approached menstruators who met the inclusion criteria and expressed their willingness to participate in the study. The researcher introduced herself and provided a brief explanation of the study's subject area. Interviews were conducted with 400 menstruators. It is worth noting that some participants mentioned sociocultural factors as a reason for their exclusion or refusal to participate. For example, some participants mentioned that their husbands or mothers-in-law disagreed with their involvement in the study. These exclusions and refusals were carefully documented and taken into consideration during the data collection process to ensure the integrity and validity of the study.

Recognizing the influence of sociocultural factors in participant recruitment and retention, the researcher acknowledged and respected these factors. Efforts were made to address any concerns or hesitations expressed by potential participants and their families. Open communication and transparent discussions were encouraged to build trust and ensure the voluntary nature of participation.

By documenting and considering these exclusions and refusals, the study aimed to account for the diverse perspectives and circumstances of potential participants, enhancing the representativeness of the final sample. This approach contributes to the overall robustness and reliability of the study's findings, as it acknowledges and accommodates the multifaceted nature of participant decision-making within a sociocultural context.

The sample size of 400 menstruators was determined after accounting for these exclusions and refusals, ensuring a representative and appropriate sample for the study. Each interview session lasted approximately 30 minutes, allowing for in-depth data collection and exploration of the research topic. The interviews were conducted in a private and comfortable setting to facilitate open and honest responses from the participants. Interviews were conducted in the local language (Kurdish) and the participants' responses were recorded directly on the questionnaire sheets. The collected data underwent secure storage through the encoding of questionnaire sheets to uphold confidentiality. Comprehensive measures were taken to ensure the secure storage and management of both hard and electronic copies. Physical questionnaire sheets, constituting hard copies, were placed in a designated, locked cabinet with restricted access. Simultaneously, electronic copies were safeguarded on a password-protected server, exclusively accessible to authorized researcher. In alignment with ethical guidelines and the protection of participants' confidentiality, a well-defined data destruction process was instituted. Following the publication of the study, all hard copies will be securely shredded to prevent unauthorized access. Similarly, electronic copies stored on the password-protected server will be deleted.

## Statistical analysis

Statistical analysis was performed using SPSS version 20.0 software. Categorical variables were presented as numbers and percentages (%). Numerical data were assessed for normal distribution. The differences between groups were analyzed using independent sample t-tests for continuous variables and $\chi2$ tests for categorical variables. A significance threshold of $P < 0.05$ was set to determine statistical significance.

## Ethical considerations

The study received approval from the Ethics Committee (Reference Number: 141) and the Midwifery Scientific Committee (Registration No: 3) at the College of Nursing, Hawler

Medical University. The researcher provided detailed and comprehensive information to participants in the study, creating a thorough Participant Information Sheet that outlines crucial aspects such as the study's objectives, research approach, potential risks, and benefits. This information was thoughtfully translated into the local language (Kurdish) to ensure clarity and accessibility. During the initial engagement with participants, the researcher prioritized transparent communication. The emphasis was placed on the pivotal role of consent, and the contents of the Participant Information Sheet were meticulously explained. This approach guarantees that participants possess full awareness of the implications of their involvement and empowers them to ask questions and seek clarifications as needed. Through this active dialogue, participants can express any concerns they might have about the study's procedures.

All individuals who menstruate and have the inclusion criteria were involved in the study, understanding that their participation was entirely voluntary and that they retained the autonomy to withdraw from the study at any point without encountering adverse consequences. They were also assured that any collected data would be used solely for the study's purpose and treated with confidentiality. Verbal consent was obtained from all menstruators who chose to participate in the study. The researcher invited them to provide their consent verbally, opting for verbal consent due to its greater acceptance in our society. Importantly, the researcher made it clear and ensures that all information remains confidential, safeguarding the privacy of the participants.

## Results

The study sample consisted of 400 participants who met the inclusion criteria. The sample does not contain who many identify as transgender men, or individuals of other gender identities. The average age of the participants was 28.35 years, with a standard deviation of 6.298. A significant proportion of the participants (24.3%) had completed primary schooling as their highest level of education, suggesting a predominantly basic educational background. The majority of the sample (37.3%) were unemployed housewives, indicating a high rate of non-participation in formal employment. Additionally, a large majority (68.3%) of the participants were married. In terms of the timing of menarche, more than half (50.5%) of the participants experienced their first menstrual cycle between the ages of 13- and 14-years Table 1.

The present study found that 63.8% of the participants had been uninfected with COVID-19. In terms of vaccination, the majority of the study sample received the Pfizer-BioNTech vaccine (76.5%), followed by Sinopharm (10.8%), Johnson & Johnson's Janssen (8.5%), and AstraZeneca (4.0%). All participants (100%) received two doses of the COVID-19 vaccine, and the majority of them (79.5%) had a one-month interval between the doses Table 2.

A majority of the participants reported changes in their menstrual cycle following vaccination. Specifically, 71.8% of the study sample mentioned experiencing irregular menstrual cycles compared to before COVID-19 vaccination. Before vaccination, 38.8% and 55.8% of the participants had a five and six-days bleeding duration(period), respectively. However, after COVID-19 vaccination, these figures decreased to 29.3% and 5.5%, respectively. Conversely, 36.5% and 20.8% of the study sample reported an increase in their bleeding duration(period) to seven and eight days, respectively, following vaccination.

Furthermore, a majority of the participants (72.3%) reported changes in the amount of blood loss during their menstrual cycle. Specifically, they mentioned a shift from light or moderate menstrual flow before vaccination to heavy flow with clots or flooding after vaccination. Additionally, 44.8% of the participants noted a change in the duration between two consecutive periods, with the interval decreasing from the normal duration between the periods to less than 21 days Table 3.

**Table 1. Socio-demographic characteristics of the study sample (n = 400).**

| Variables | | n (%) |
|---|---|---|
| Age | Mean + SD 28.35 (± 6.298) | |
| **Educational level** | | |
| Illiterate | | 49 (12.3) |
| Primary | | 97 (24.3) |
| Secondary | | 84 (21.0) |
| Institute | | 55 (13.8) |
| College | | 80 (20.0) |
| High education | | 35 (8.80) |
| **Occupational** | | |
| Formal employment | | 71 (17.8) |
| Self-employed | | 100 (25.0) |
| Unemployed | | 149 (37.3) |
| Student | | 80 (20.0) |
| **Marital status** | | |
| Married | | 273 (68.3) |
| Unmarried | | 127 (31.8) |
| 1$^{st}$ Menstrual age/ Years | | |
| 10–12 | | 198 (49.5) |
| 13–14 | | 202 (50.5) |

The findings from the present study indicate that there were notable changes in menstrual cycle signs and symptoms before and after COVID-19 vaccination. Specifically, 75.8% of the participants reported experiencing dysmenorrhea (menstrual pain), which increased to 93% after vaccination. Similarly, the rate of lower back pain increased from 53.5% before vaccination to 87.5% after vaccination. Furthermore, the percentage of participants experiencing nausea and vomiting rose from 6.8% before vaccination to 24.5% after vaccination. In contrast, no significant changes were observed in the rate of genital rash, with 10% reported both before and after vaccination. The percentage of participants experiencing vaginal irritation decreased from 11% before vaccination to 6.5% after vaccination. Likewise, the percentage of participants reporting genital redness or inflammation remained consistent at 3% before and after COVID-19 vaccination Table 4.

A statistically significant change in the menstrual cycle characteristics, as well as the signs and symptoms of the menstrual cycle, before and after COVID-19 vaccination p-value of

**Table 2. Information regarding the infected by COVID-19 infection and received COVID -19 Vaccine (n = 400).**

| Variables | n (%) |
|---|---|
| Infected by Covid -19 infection | 145 (36.3) |
| Uninfected by Covid -19 infection | 255 (63.8) |
| **Types of Covid-19 vaccine** | |
| Pfizer-BioNTech. | 307 (76.8) |
| AstraZeneca | 16 (4.0) |
| Johnson & Johnson's Janssen | 34 (8.5) |
| Sinopharm | 43 (10.8) |
| **Duration between Dosages** | |
| 1month | 318 (79.5) |
| 2months | 82 (20.5) |

**Table 3. Provides information on the menstrual cycle characteristics before and after COVID-19 vaccination among the study sample (n = 400).**

| Variables | Before Vaccination n (%) | After Vaccination n (%) | p—values |
|---|---|---|---|
| **Regularity of menstrual cycle** | | | |
| Yes | 400(100.0) | 113(28.2) | |
| No | 0.0 (0.00) | 28(71.8) | |
| **Days of bleeding (period)** | | | 0.000 |
| 4 | 0 (0) | 11(2.8) | |
| 5 | 155 (38.8) | 117(29.3) | |
| 6 | 223 (55.8) | 2 (0.5) | |
| 7 | 22 (5.5) | 146 (36.5) | |
| 8 | 0 (0.0) | 83 (20.8) | |
| 9 | 0 (0.0) | 21 (5.3) | |
| 10 | 0 (0.0) | 14 (3.5) | |
| 11 | 0 (0.0) | 6 (1.5) | |
| **Heavy menstrual flow** | | | 0.000 |
| light | 125 (31.3) | 13(3.3) | |
| Moderate | 275 (68.8) | 98 (24.5) | |
| Heavy (clots/flooding) | 0 (0.0) | 289 (72.3) | |
| **Duration between periods** | | | 0.000 |
| Less than 21 days | 19 (4.8) | 179 (44.8) | |
| 22–24 days | 5 (1.3) | 74 (18.5) | |
| 25–28 days | 253 (63.3) | 81 (20.3) | |
| 29–32 days | 87(21.8) | 15 (3.8) | |
| 33–35 days | 36 (9.0) | 30 (7.5) | |
| Too irregular to say | 0 (0.0) | 21 (5.3) | |

0.000. Specifically, there were significant alterations observed in the following menstrual cycle characteristics: days of bleeding, heavy menstrual flow, and the duration between two consecutive periods. These changes indicate a meaningful impact of COVID-19 vaccination on the duration and intensity of menstrual bleeding. Furthermore, the signs and symptoms of the menstrual cycle also demonstrated significant changes. These changes include dysmenorrhea (menstrual pain), lower back pain, and nausea and vomiting. The statistically significant findings highlight the impact of COVID-19 vaccination on these menstrual cycle signs and symptoms Tables 3 and 4.

The highest percentage (43.5%) of the study sample were reported returning to their regular menstrual cycle three months after receiving the COVID-19 vaccination. This suggests that for

**Table 4. Changes in menstrual cycle signs and symptoms before and after COVID-19 vaccination n = 400.**

| Signs and Symptoms | Before Vaccination | | After Vaccination | | P- Value |
|---|---|---|---|---|---|
| | Yes n (%) | No n (%) | Yes n (%) | No n (%) | |
| Dysmenorrhea | 303 (75.8) | 97 (24.3) | 372 (93.0) | 28 (7.0) | 0.000 |
| Lower back pain | 214 (53.5) | 186 (46.5) | 350 (87.5) | 50 (12.5) | 0.000 |
| Nausea & Vomiting | 27 (6.8) | 373 (93.3) | 98 (24.5) | 302 (75.5) | 0.000 |
| Genital rash | 40 (10.0) | 360 (90.0) | 40 (10.0) | 360 (90.0) | 0.297 |
| Vaginal irritation | 44 (11.0) | 356 (89.0) | 25 (6.3) | 375 (93.8) | 0.047 |
| Genital redness or inflammation | 12 (3.0) | 388 (97.0) | 12 (3.0) | 388 (97.0) | 0.312 |

**Table 5. Time to return to normal menstrual cycle after COVID-19 vaccination n = 400.**

| Variables | n (%) |
|---|---|
| Return the period to normal state / Months | |
| 2 | 70 (17.5) |
| 3 | 174(43.5) |
| Till now | 43(10.8) |
| Regular | 113(28.3) |
| Total | 400(100.0) |

a significant proportion of individuals, it took approximately three months for their menstrual cycle to normalize following vaccination Table 5.

## Discussion

Understanding the potential impact of COVID-19 vaccination on menstrual cycles within Iraqi society is crucial. While specific data related to Iraq is limited, studies from other countries have emphasized that some individuals may experience changes in their menstrual cycles following COVID-19 vaccination. These changes can manifest as irregularities in cycle length, alterations in bleeding duration, variations in blood loss volume, and shifts in cycle regularity [10,11,15,16]. Given the scarcity of research on this topic in the Kurdistan Region of Iraq, it becomes imperative to investigate and gather information to address this knowledge gap, providing valuable insights into the connection between women's health and COVID-19 vaccination.

Caroline Criado-Perez's 'Invisible Women' draws attention to the oversight of menstruators in public health research, particularly in trial studies. In aligning our findings with Criado-Perez's observations, we underscore the continued need for a more inclusive approach to reproductive health in the context of public health initiatives [5].

The study not only contributes to the growing body of research on COVID-19 vaccination but also serves as a case study within the broader discourse on menstruation in public health. The observed changes in menstrual patterns among the study participants echo the concerns raised in 'Invisible Women' about the systemic neglect of women's health issues in medical research. Excluding women or menstruators from research studies poses significant problems and has wide-ranging implications. First and foremost, women make up half of the population, and menstruation is a natural and significant aspect of many individuals' lives. By excluding them from studies, not only ignore the health needs and experiences of a substantial portion of the population but also risk overlooking potential differences in how medical interventions, such as COVID-19 vaccination, affect different genders.

Furthermore, the statistically significant changes in menstrual cycle characteristics and associated signs and symptoms underscore the need for ongoing monitoring and research in this area. While the exact mechanisms linking COVID-19 vaccination to menstrual changes remain unclear, our findings contribute to the growing body of evidence suggesting a potential association.

The findings of this study reveal that the most commonly used COVID-19 vaccines in Iraq are Pfizer-BioNTech, Sinopharm, Johnson & Johnson's Janssen, and AstraZeneca, in descending order. Notably, Pfizer-BioNTech holds significant importance as it was the first COVID-19 vaccine to receive Emergency Use Authorization (EUA) from the Food and Drug Administration (FDA) in December 2020. This authorization was granted based on its demonstrated effectiveness in preventing COVID-19 symptoms. Subsequently, Pfizer-BioNTech became the

first COVID-19 vaccine to receive full FDA approval in August 2021 for individuals aged 16 years and older [17]. Supporting the prominence of the Pfizer vaccine in Iraq, a study by Hamza et al. in 2022 found that Pfizer and BioNTech vaccines were the most widely used across 10 countries, including Saudi Arabia, UAE, Qatar, Kuwait, Oman, Bahrain, Jordan, Lebanon, Tunisia, and Iraq [18]. These findings, as shown in Table 2, underscore the high vaccination rate and adherence to recommended dosing intervals, serving as positive indicators of the study's vaccination efforts.

The study's results, presented in Table 3, highlight a prevalent trend: a significant number of participants reported distinct changes in their menstrual cycle characteristics after COVID-19 vaccination, particularly after completing both vaccine doses. This underscores the substantial impact of COVID-19 vaccination on menstrual cycles, evidenced by alterations in cycle regularity, bleeding duration, blood loss volume, and intermenstrual interval. These findings align with a systematic review by Nazir et al. (2022), suggesting a connection between the second vaccine dose and menstrual irregularities [10]. Before vaccination, participants exhibited regular menstrual cycles at a 100% rate. However, our study revealed noticeable changes post-vaccination, including an extension of bleeding duration from 6 to 7–8 days. A similar consistency was observed in the study by Amer et al. (2022), encompassing several Arabic countries, which reported up to an 8-day variation in menstrual duration [9]. Significantly, alongside prolonged bleeding, over 70% of participants experienced heavy bleeding characterized by excessive flow and the presence of clots. This aligns with a Saudi Arabian study that reported analogous post-vaccination menstrual changes, including increased bleeding [19].

Furthermore, a decrease in the intermenstrual interval post-vaccination was observed, with most participants transitioning from a normal interval of 25–28 days to less than 21 days. This change corresponds with findings from a Saudi Arabian study indicating altered cycle length and shorter-than-usual cycles [19].

Shifting the focus to subjective experiences, the study examined pre- and post-vaccination menstrual cycle symptoms. Painful menstrual cramps (dysmenorrhea), lower back pain, nausea, vomiting, genital rash, vaginal irritation, and genital redness were among the investigated symptoms. Notably, the study identified an increase in the severity of dysmenorrhea after vaccination, along with elevated reports of lower back pain and nausea/vomiting. Statistical analysis confirmed significant differences in symptom occurrence before and after vaccination, as demonstrated in Wang et al.'s study (2022) [19,20].

The documented changes in menstrual cycle characteristics following COVID-19 vaccination have raised crucial considerations regarding both individual menstruators' health and public health on a broader scale. These findings suggest that COVID-19 vaccination can have effects beyond its immediate immunological impacts, potentially influencing hormonal regulation and reproductive health. From an individual health perspective, altered menstrual cycles could lead to discomfort, inconvenience, and anxiety for menstruators. Prolonged bleeding duration (days of bleeding), heavy bleeding, and shorter intermenstrual intervals may result in physical discomfort and emotional distress. Menstruators who rely on consistent menstrual patterns to monitor reproductive health or plan activities may also face challenges due to these changes.

Moreover, the investigation unveiled that irregular menstrual signs and symptoms resolved within three months following COVID-19 vaccination for 43.5% of the participants, in line with findings by Wang et al. regarding delayed resolution [20]. Notably, a significant number of participants reported reverting to their typical menstrual cycle approximately three months after receiving the COVID-19 vaccination. This observation raises intriguing questions about the duration of these alterations, emphasizing the importance of longitudinal studies in monitoring the persistence or resolution of modified menstrual patterns over time.

However, it is important to address the study's limitations. The cross-sectional design impedes establishing causal relationships between COVID-19 vaccination and menstrual changes. The potential for recall bias due to the participation of individuals with pre-existing menstrual disorders is acknowledged. Additionally, while the study sheds light on a specific population sample, its applicability to broader regions may be limited.

## Conclusion

In conclusion, the study underscores the significance of COVID-19 vaccination's impact on menstrual cycles within the Iraqi society. By revealing distinct changes in cycle characteristics and symptomatology, the study findings contribute to an expanding body of evidence supporting the connection between vaccination and menstrual alterations that includes: dysmenorrhea, lower back pain, and nausea/vomiting. Addressing the study's limitations, future research could adopt longitudinal designs and encompass larger, diverse samples to gain a more comprehensive understanding of these dynamics. Ultimately, advancing this knowledge supports informed healthcare decisions for menstruators navigating the intersection of vaccination and menstrual health.

## Supporting information

**S1 File. Questionnaire (English language).**
(PDF)

## Author Contributions

**Conceptualization:** Warda Hassan Abdullah.

**Data curation:** Warda Hassan Abdullah.

**Funding acquisition:** Warda Hassan Abdullah.

**Methodology:** Warda Hassan Abdullah.

**Project administration:** Warda Hassan Abdullah.

**Supervision:** Warda Hassan Abdullah.

**Writing – original draft:** Warda Hassan Abdullah.

**Writing – review & editing:** Warda Hassan Abdullah.

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
