## [Decision Letter · Decision Letter 0]

24 Mar 2023

PGPH-D-23-00059

Menstrual Changes after COVID-19 Vaccination among Women of Reproductive Age: A Cross- Sectional Study from Erbil City, Iraq

Dear Dr. Abdullah,

Thank you for submitting your manuscript to PLOS Global Public Health. After careful consideration, we feel that it has merit but does not fully meet PLOS Global Public Health’s publication criteria as it currently stands. Therefore, we invite you to submit a revised version of the manuscript that addresses the points raised during the review process.

Both reviewers raised significant concerns about the work. Please address all the comments point-by-point.

We look forward to receiving your revised manuscript.

Kind regards,

Jianhong Zhou

Staff Editor

Journal Requirements:

1. In the ethics statement in the Methods, you have specified that verbal consent was obtained. Please provide additional details regarding how this consent was documented and witnessed, and state whether this was approved by the IRB.

2. We ask that a manuscript source file is provided at Revision. Please upload your manuscript file as a .doc, .docx, .rtf or .tex.

4. Please amend your Data Availability Statement and indicate where the data may be found.

Additional Editor Comments (if provided):

Reviewers' comments:

Reviewer's Responses to Questions

**Comments to the Author**

1. Does this manuscript meet PLOS Global Public Health’s publication criteria? Is the manuscript technically sound, and do the data support the conclusions? The manuscript must describe methodologically and ethically rigorous research with conclusions that are appropriately drawn based on the data presented.

Reviewer #1: Yes

Reviewer #2: No

2. Has the statistical analysis been performed appropriately and rigorously?

Reviewer #1: I don't know

Reviewer #2: Yes

3. Have the authors made all data underlying the findings in their manuscript fully available (please refer to the Data Availability Statement at the start of the manuscript PDF file)?

Reviewer #1: Yes

Reviewer #2: No

4. Is the manuscript presented in an intelligible fashion and written in standard English?

Reviewer #1: No

Reviewer #2: No

5. Review Comments to the Author

Reviewer #1: General

• With recognition of those with gender-nonconforming identities such as transgender men and some non-binary persons using the term women to indicate those that menstruate is not inclusive. Please instead use the term ‘menstruator’ of reproductive age. This could even be turned into an acronym MORA. In the new definition for Menstrual Health this term is used along with a justification of using gender inclusive terms if you’d like a reference https://www.tandfonline.com/doi/full/10.1080/26410397.2021.1911618

• The English is not perfect throughout the document – I’m not sure how much this matters as I can still mostly understand the meaning, but other non-native English speakers may struggle. It would take too long to list every error – perhaps just give it another read through, or ask a colleague to check it? A general edit to make is that you often repeat the same word in one sentence unnecessarily e.g. women or disease in the introduction. I’m unsure if this journal offers proofreading for non-native English writers?

• A strong narrative would be to point out that women’s health is often neglected from data collection, which is why they are often misdiagnosed or not diagnosed at all – symptoms of autism and heart attacks are based on the male experience. Ref – book, Invisbile Women, Caroline Criado Perez. So I imagine that this was the case for testing the COVID vaccine too – that and the invisibility of menstrual health. I think this would strengthen the paper.

Specific

• ‘can disrupt the luteal phase, cause ovulation to stop, and lead to scanty infrequent periods, or oligomenorrhea’ – it sounds like they experience one or the other rather than oligomenorrhea is the medical term for what you’ve described. Rewrite for clarity.

• ‘because anxiety is associated with higher menstrual scores’ – what are menstrual scores?

• ‘menstruating people’ – use ‘menstruators’

• ‘in their study was funded that changes in menstrual volume were’ – do you mean found?

• ‘Finally, the investigates in their study’ – I’m not sure what investigates are

• ‘In another hand’ – should be ‘on the other hand’

• Methods – why was a population proportion of 50% chosen?

• What are the professional backgrounds of the ‘experts’?

• In the sample size you propose to interview 400 people and then later you say you interviewed 500 women?

• Results – ‘Women who participated in the study were in reproductive age’ you don’t need to repeat this you’ve already stated in methods

• ‘Regarding to socio-demographic data the (Table 1) shows that highest percentage (57.1%) (58.9%) of the study sample’ – Sorry I don’t understand this sentence

• ‘Above twenty (24.3%) of the participants had a primary schooling while’ – above 20 what?

• ‘Maternal status’ – you mean marital status

• Results – why have you chosen to highlight some stats but not others? The writing in the results should add something to the data that’s in the table, not just repeat it

• Could the results of the table 3 impacts on menstruation be instead presented in some form of visual graph so that the reader can really see the differences before and after covid/the vaccine. It’s harder to digest the information when it’s just numbers in a table

• ‘and about seventy (36.5%, 20.8%, 5.3%,3.5%, and 1.5% respectively)’ - can you make it clearer to the reader where these percentages come from – especially since they’re from the table after the text

• Dysmenorrhea – give a definition for this

• Did demographic data like age have any impact on the effects of the vaccine on their menstruation?

• ‘Future studies are necessary’ – what would be the purpose of future studies?

• I’d like to see more explanations/discussion on why this was a shortcoming of those developing the vaccine i.e. what have been the detrimental impacts of these changes in menstruator’s periods, how this could be avoided in the future, and how scientists and healthcare professionals can warn people of these side effects

• The section on cycle length in the discussion is very lengthy and doesn’t give much insight

Reviewer #2: This paper brings a global view on the association between the Covid vaccine and women's menstrual cycles. The focus on Iraqi women is an important contribution. However, there are some major issues with the paper:

General:

1) The entire article has to be reviewed for grammar, typos, etc. The lack of attention to detail makes it very sloppy and make it seem as if the author does not care about the work. It's a sign of respect (for the work, the participants, the reviewers, and the reader) to make sure Covid is not spelled "Coved"

2) It's strange that there is only one author on this work, considering the magnitude of the endeavor and the involvement of several centers and experts. This reviewer would have expected a multi-disciplinary authorship team, with a gender representation of scientists and contributors.

3)The paper could benefit from commentary on culture and menstruation in this part of the world; and why this work is important in the context of the Covid-19 pandemic and vaccines.

Specifics:

Introduction:

1)this sentence "However, research hasn’t directly linked these changes to the COVID-19 vaccines. In

fact, research into the way vaccines can impact menstruation is generally lacking" is immediately followed by this sentence, which puts the two at contradictory ends of the spectrum "The Medicines and Healthcare Products Regulatory Agency (MHRA) in UK

reported that between December 8, 2020 and September 8, 2021 there were 34,633

incidents of menstrual and vaginal bleeding reactions reported to them in relation to a

COVID-19 vaccine in Great Britain"

There is also AMPLE research on the Covid vaccines and female menstruation that we implore the author to review; including the Apple Women's Study. Infact, the author mentions this work in the Discussion. So why are they saying there is limited data on Covid vaccines and menstruation? This is not true, nor is it strong justification of the study.

2)With that said, a stronger case should be made for the focus on Iraqi women rather than the view that there is, globally, limited research on Covid and menstrual changes because that is not true and this entire paragraph needs to be reworked to focus on the study's population, the fact that the cross-sectional design does not get at the "short and long term" effects of the vaccine, the focus on Covid infection history/not, etc.

"The researcher thought society need this study because some women have

reported changes in their menstruation after receiving the COVID-19 vaccine,

including changes in duration, flow, and accompanying symptoms such as pain. In

another hand, the importance of the study as more people are vaccinated for COVID19, it is possible to gain a better understanding of the short- and long-term effects of

the vaccines. Scientific evidence could also help unvaccinated people understand

what, if any, menstruation-related side effects to expect from a COVID-19 vaccine.

Finally, the aim of the study to identify if have a connection between vaccination by

(COVID-19 vaccine) and changes in the menstrual cycle of women of reproductive"

Methods:

1) is this true for Iraq or for ALL places in the world? If the latter, it's not a true statement.

"The researcher selected the primary health care centres as the setting of the study because women of reproductive

age visit primary health care centres for health services."

2)What does this mean? "contaminated and uncontaminated COVID19 "

3) why this inclusion criteria? What is the biological premise? "received the COVID-19 vaccine no

longer than 3 mounts included in the study"

4) "But, for reliability estimated 400 women of

reproductive age were included in the study" what reliability?

5) Some details on why participants refused to participate in the study will be interesting...see comment on culture and menstruation.

6) Grammar and sentence structure in sentences below. Also, aren't categorical variables also "numeric"? do you mean "continuous variables?" instead of numerical data?

"Categorical variables in numbers and percentages (%). Numerical data were analyzed

for normal distribution."

Results:

1) Recommend removing this phrase and getting to the point faster:

"Women who participated in the study were in reproductive age. All of them

received Covid-19 vaccine and their menstrual cycle were regular before received

Covid-19 vaccine. The study sample were free from hormonal disease and don't take

hormonal therapy."

2) Having trouble understanding this sentence "Regarding to socio-demographic data the (Table 1) shows that

highest percentage (57.1%) (58.9%) of the study sample"and this too "Concerning

COVID-19 infection, 63.8% of participants had contaminated. "

3)This is great data; and would be good to see if any one of these vaccines was more associated with the changes reported versus others; especially since the discussion begins with a sentence about the vaccine types:

"The majority of the

study sample received Pfizer-BioNTech, Sinopharm, Johnson & Johnson's Janssen

and AstraZeneca (76.5%, 10.8%, 8.5 and 4.0%, respectively), and all of them (100%)

received two doses of Covid- 19 vaccine and the 79.5% of them was one month

between the dosages "

4) SEVERAL typos on all the tables and headers and perhaps a number of these tables can be combined. Not very rigorous; merely descriptive and self-reported data.

5) the survey instrument is not shown, and the methods for creating it do not seem that rigorous...

6) Why was it important to collect data on vaginal redness/inflammation. The justification for this, as it relates to the vaccine is not provided and/or discussed anywhere...Interesting variable to collect though!

Discussion:

1) In the introduction author mentions "There is limited data on Covid vaccination and menstruation" but then have this paragraph in the discussion. So the premise of the research is invalidated...the introduction and justification need re-working as the current study is not adding anything new to the literature.

A systematic review study conducted by Nazir et al 2022, who found after

receiving the second dose of the COVID-19 vaccine was a predictor of menstrual

problems, after receiving the COVID-19 vaccine, women reported a significantly

increased in the duration of their menstrual cycles compare to pre- COVID-19 vaccine

status(12) in another hand, the finding of the current study is consistent with an Arabic

Countries (Saudi Arabia, Egypt, Syria, Libya, and Sudan) study conducted by Amer

et al. who have reported that the length of a menstrual duration varies by up to 8 days

it is abnormal(10, 13).

2) The recall bias i)does not only affect those with menstrual irregularities but has to do with the study design and ii)the analyses should include a table with menstrual irregularities excluded.

3) The discussion is not coherent and needs reworking. Again, a bit sloppy.

6. PLOS authors have the option to publish the peer review history of their article (what does this mean?). If published, this will include your full peer review and any attached files.

**Do you want your identity to be public for this peer review?** For information about this choice, including consent withdrawal, please see our Privacy Policy.

Reviewer #1: **Yes: **Georgia Hales

Reviewer #2: No

---

## [Decision Letter · Decision Letter 1]

27 Jul 2023

PGPH-D-23-00059R1

Menstrual Changes after COVID-19 Vaccination among Menstruator of Reproductive Age: A Cross- Sectional Study from Erbil City, Iraq

Dear Dr. Warda Hassan Abdullah,

Thank you for submitting your manuscript to PLOS Global Public Health. After careful consideration, we feel that it has merit but does not fully meet PLOS Global Public Health’s publication criteria as it currently stands. Therefore, we invite you to submit a revised version of the manuscript that addresses the points raised during the review process.

Please address the reviewers' comments and submit your revised manuscript by 25 August 2023. If you will need more time than this to complete your revisions, please reply to this message or contact the journal office at globalpubhealth@plos.org. Please include the following items when submitting your revised manuscript:

We look forward to receiving your revised manuscript.

Kind regards,

Md. Nazmul Huda, PhD

Academic Editor

Journal Requirements:

1. In the ethics statement in the Methods, you have specified that verbal consent was obtained. Please provide additional details regarding how this consent was documented and witnessed, and state whether this was approved by the IRB.

Additional Editor Comments (if provided):

Reviewers' comments:

Reviewer's Responses to Questions

**Comments to the Author**

1. If the authors have adequately addressed your comments raised in a previous round of review and you feel that this manuscript is now acceptable for publication, you may indicate that here to bypass the “Comments to the Author” section, enter your conflict of interest statement in the “Confidential to Editor” section, and submit your "Accept" recommendation.

Reviewer #1: All comments have been addressed

Reviewer #3: (No Response)

2. Does this manuscript meet PLOS Global Public Health’s publication criteria? Is the manuscript technically sound, and do the data support the conclusions? The manuscript must describe methodologically and ethically rigorous research with conclusions that are appropriately drawn based on the data presented.

Reviewer #1: Yes

Reviewer #3: No

3. Has the statistical analysis been performed appropriately and rigorously?

Reviewer #1: Yes

Reviewer #3: (No Response)

4. Have the authors made all data underlying the findings in their manuscript fully available (please refer to the Data Availability Statement at the start of the manuscript PDF file)?

Reviewer #1: Yes

Reviewer #3: (No Response)

5. Is the manuscript presented in an intelligible fashion and written in standard English?

Reviewer #1: Yes

Reviewer #3: No

6. Review Comments to the Author

Reviewer #1: Thank you for your revised version and taking our comments into consideration. You’ve clearly put in a lot of effort and the paper is looking great. So important to shed light on this topic in different contexts – I think it adds a lot of value to the discourse. I have just a few more changes that could be made if you agree. It was a pleasure to read and review.

Review comments

• Some passive voice – consider changing to active voice e.g. the authors utilized a structured questionnaire rather than A structured questionnaire was utilized

• Maybe just specify that the menstruators are of reproductive age just once, and then continue to refer to them just as menstruators as ‘menstruators of reproductive age’ is a bit clunky to keep repeating

• Perhaps you want to explain the use of the term 'menstruator' somewhere in case the reader is not familiar – in that gender diverse persons may or may not menstruate so the term ‘woman’ is not inclusive of these other persons e.g. trans men

• Would another key word be COVID-19 vaccine?

• Title – should be menstruators (plural)

• I still feel that there are a few areas of repetition – for example in your introduction you repeat the aim of the study twice, and then again in the first line of your method, and then in the discussion – maybe limit this to just twice throughout the paper

• You explain about using a slightly larger sample size than 400 – can you please state that it was 500 when explaining this.

• I would explain that dysmenorrhea is specifically painful menstrual cramps, rather than just menstrual pain. Also you already explain its meaning once, no need to repeat.

• I know it’s extra work but perhaps some of the results could be visually represented by a graph – it would make it easier for the reader to see the differences before and after the vaccine. The tables are great but it’s a lot of numbers to digest

• I find the discussion a little bland – it is mainly presenting the results again but comparing them with other studies. I love this point in your conclusion – ‘’The menstrual cycle is an important indicator of menstruator health, and disruptions in its regularity can signal underlying conditions. Therefore, it is essential to increase awareness among healthcare professionals and menstruator about potential menstrual abnormalities following Covid19 vaccination.’’ Perhaps you could introduce and expand on this a little bit more in the discussion – and get more into why are these findings important. From my view they’re important because (a) it shows the lack of consideration of menstruators and menstruation in public health, (b) it may have caused unnecessary physical discomfort and stress and (c) your concluding point that we should be aware of these side-effects so people can prepare for them and understand what’s happening in their bodies.

Reviewer #3: 19072023

Reviewer’s comments

This paper aims to establish an association between Covid vaccination and menstruation. Selecting an Iraqi city for data collection is a valuable consieration. Unfortunately, there are some major issues in the research conduct that restricted generating a meaningful and reliable evidence. Therefore, it is recommended to re-conduct the research and not to publish the findings. The issues are as follows:

• Literature review: It is unclear what data bases were searched for the literature review and how this search was unbiased or not selective. There is only one researcher involved and lack of contribution from a subject librarian may enhance the chance of information bias.

• Method:

a. The study failed to describe how the experts were selected, how their opinion was collected in the development of the data collection tool.

b. For the data collection, in-depth interview method was used but there is no indication of what language was used, whether it was recorded and transcribed, where it was stored, how and when it was anonymised. Also, there was no information around the nature of the data, was it all qualitative or there are some quantitative data as well? It was also absent in the description of data collection method if there were any quantifiable data around the degree of changes in menstruation.

c. There was no mention of how the data had been managed in terms of storing and destroying. It was unclear where the hard and soft copies of data were stored, who were responsible for their maintenance and when they should be destroyed.

d. There was no mention of GDPR.

e. There was no qualitative data analysis and no explanation around what happened with the qualitative data collected by conducting interviews.

f. Recruitment: a convenience sampling was used but there was inadequate description of how the participants were recruited, what were their selection criteria, who recruited them, whether there personal data were accessed and if so, then who accessed them, whether there was any permission from an appropriate authority around that etc.

Also it was unclear how 500 participants were interviewed and how 10 of them did not agree to share information.

Lack of these details of the methodology it is insufficient to allow the experiments to be reproduced

• Ethics:

- It is not evident that the study is ethically sound. In the first meeting with the potential participants, the author/ researcher introduced the study to them. There was no Participants Information Sheet shared with the potential participants. As a result there was no scope for the participants to revisit the information about the study aim, expectation, time commitment, risks and benefit, impact etc. and decide on their participation.

- Verbal consent was obtained from the participants and there is no information on whether it has been recorded, if so how and whether this was witnessed.

• Results: The manuscript has serious concerns around the methodology of data collection, data management and data analyses. As a result, the derived conclusions and results are not reliable. Moreover, absence of sub-group analysis made the claims much weaker.

7. PLOS authors have the option to publish the peer review history of their article (what does this mean?). If published, this will include your full peer review and any attached files.

**Do you want your identity to be public for this peer review?** For information about this choice, including consent withdrawal, please see our Privacy Policy.

Reviewer #1: **Yes: **Georgia Hales

Reviewer #3: No

---

## [Decision Letter · Decision Letter 2]

16 Oct 2023

PGPH-D-23-00059R2

Menstrual Changes after COVID-19 Vaccination among Menstruators of Reproductive Age: A Cross- Sectional Study from Erbil City, Iraq

Dear Dr. Abdullah,

Thank you for submitting your manuscript to PLOS Global Public Health. After careful consideration, we feel that it has merit but does not fully meet PLOS Global Public Health’s publication criteria as it currently stands. Therefore, we invite you to submit a revised version of the manuscript that addresses the points raised during the review process.

The manuscript has been evaluated by two reviewers, and their comments are available below.

The reviewers have raised a number of major concerns. They feel the manuscript should outline a clearly-defined research question, and they request improvements to the reporting of methodological aspects of the study, for example, regarding the exclusion criteria and more information on how the data collection was completed. The reviewers also note concerns about the statistical analyses presented and request re-analyses be completed.

Could you please carefully revise the manuscript to address all comments raised?

We look forward to receiving your revised manuscript.

Kind regards,

Avanti Dey, PhD

Staff Editor

Journal Requirements:

Additional Editor Comments (if provided):

Reviewers' comments:

Reviewer's Responses to Questions

**Comments to the Author**

1. If the authors have adequately addressed your comments raised in a previous round of review and you feel that this manuscript is now acceptable for publication, you may indicate that here to bypass the “Comments to the Author” section, enter your conflict of interest statement in the “Confidential to Editor” section, and submit your "Accept" recommendation.

Reviewer #1: (No Response)

Reviewer #4: (No Response)

2. Does this manuscript meet PLOS Global Public Health’s publication criteria? Is the manuscript technically sound, and do the data support the conclusions? The manuscript must describe methodologically and ethically rigorous research with conclusions that are appropriately drawn based on the data presented.

Reviewer #1: Partly

Reviewer #4: Yes

3. Has the statistical analysis been performed appropriately and rigorously?

Reviewer #1: I don't know

Reviewer #4: Yes

4. Have the authors made all data underlying the findings in their manuscript fully available (please refer to the Data Availability Statement at the start of the manuscript PDF file)?

Reviewer #1: Yes

Reviewer #4: Yes

5. Is the manuscript presented in an intelligible fashion and written in standard English?

Reviewer #1: Yes

Reviewer #4: Yes

6. Review Comments to the Author

Reviewer #1: The researcher has largely addressed my comments. I just have a few more on methods, ethics and the discussion.

Method and tool data collection

Line 201 – where was the data securely stored? You go on to explain this in data management but you can indicate it here/make note that you will go on to explain in the following section. How was it anonymized?

Ethical considerations

Line 230 – who witnessed the verbal consent being given aside from the researcher?

Line 243 – when will the copies be shredded? What is meant by study’s completion – once the paper is published?

Discussion

The researcher did not address the comment of adding into the discussion that the results demonstrate a the lack of consideration of menstruators and menstruation in public health, namely not using enough menstruators in drug trial studies (reference book ‘Invisible Women by Caroline Criado-Perez) or considering menstruation as a public health concern.

Reviewer #4: In this study, the authors investigated the effect of covid 19 vaccination on menstrual cycle. The research question is very popular nowadays because of many unknowns abot covid 19 vaccination and the authors ver well explained the need of this research especillay in Iraq. I have a few suggestions and questions;

1. The authors used the term “menstruator” for women menstrating regulary but “Menstruators is an inclusive term for all people that menstruate, which includes cisgender women, transgender men, non-binary and intersex people” and this term is natural for gender. I suggest the authors to add a sentence to material method or result section if there is transgender men or intersex people among the participants.

2. Is there a difference in terms of menstrual irregularities between theinfected and uninfected groups?

3. Is there a need of table 5? It would be easy to follow the tables if the authors could add a p value for the difference between groups for every sign and symptoms in table 4 and table 3

4. It is unclear whether the interviewed women took anticoagulants during and after their infection, as this could affect the duration of menstruation and the amount of blood lost. This question should be included in the questionnaire

7. PLOS authors have the option to publish the peer review history of their article (what does this mean?). If published, this will include your full peer review and any attached files.

**Do you want your identity to be public for this peer review?** For information about this choice, including consent withdrawal, please see our Privacy Policy.

Reviewer #1: No

Reviewer #4: **Yes: **Ozlem Turhan Iyidir

---

## [Decision Letter · Decision Letter 3]

18 Dec 2023

PGPH-D-23-00059R3

Menstrual Changes after COVID-19 Vaccination among Menstruators of Reproductive Age: A Cross- Sectional Study from Erbil City, Iraq

Dear Dr. Abdullah,

Thank you for submitting your manuscript to PLOS Global Public Health. After careful consideration, we feel that it has merit but does not fully meet PLOS Global Public Health’s publication criteria as it currently stands. Therefore, we invite you to submit a revised version of the manuscript that addresses the points raised during the review process.

We look forward to receiving your revised manuscript.

Kind regards,

Avanti Dey, PhD

Staff Editor

Journal Requirements:

1. In the ethics statement in the Methods, you have specified that verbal consent was obtained. Please provide additional details regarding how this consent was documented and witnessed, and state whether this was approved by the IRB.

Additional Editor Comments (if provided):

Reviewers' comments:

Reviewer's Responses to Questions

**Comments to the Author**

1. If the authors have adequately addressed your comments raised in a previous round of review and you feel that this manuscript is now acceptable for publication, you may indicate that here to bypass the “Comments to the Author” section, enter your conflict of interest statement in the “Confidential to Editor” section, and submit your "Accept" recommendation.

Reviewer #1: (No Response)

Reviewer #4: All comments have been addressed

2. Does this manuscript meet PLOS Global Public Health’s publication criteria? Is the manuscript technically sound, and do the data support the conclusions? The manuscript must describe methodologically and ethically rigorous research with conclusions that are appropriately drawn based on the data presented.

Reviewer #1: Yes

Reviewer #4: Partly

3. Has the statistical analysis been performed appropriately and rigorously?

Reviewer #1: Yes

Reviewer #4: Yes

4. Have the authors made all data underlying the findings in their manuscript fully available (please refer to the Data Availability Statement at the start of the manuscript PDF file)?

Reviewer #1: Yes

Reviewer #4: Yes

5. Is the manuscript presented in an intelligible fashion and written in standard English?

Reviewer #1: Yes

Reviewer #4: Yes

6. Review Comments to the Author

Reviewer #1: Thanks for this next draft following our comments – I can see that you’ve taken much of what I’ve said into consideration and the paper is looking much better. I still have a few minor comments to address but apart from that I’m happy with how it is. Well done.

• Introduction

o I still feel the paper is a bit repetitive in parts. Good to be clear and guide the reader but I think you state the objective too many times. For starters you could probably delete:

Introduction - ‘The primary objective of this study is to investigate the relationship between COVID19 vaccination and menstrual changes among menstruators in Iraqi society’

Discussion - ‘Menstrual Changes after COVID-19 Vaccination among Menstruators of Reproductive Age’

• General

o Still areas where passive voice is used – would be great if the whole paper was in active voice

o Sometimes you’re missing full stops

o A few grammatical mistakes dotted throughout – please revise

• Methods

o I’m still a bit confused about the sample size calculation – had you planned for 500 or 400 participants? It seems convenient that you planned 500 with the knowledge that 100 would drop out to leave you with 400. You discuss sample size multiple times throughout the methods – would be neater to just have one section on it.

• Results

o ‘more than half’ – not scientific be specific – could say majority or just actual percentage

o Duration of period - you mean full menstrual cycle length. The word period is also used to denote the days of bleeding so this could be misleading

o P value of exactly 0? Wow.

o You don’t need to explain what dysmenorrhea is more than once

o Table 4

Would it be good to give us the % change? Perhaps the P value is enough.

why haven’t you put in the P values for the last 3 symptoms?

• Discussion

o Why is it a problem that women/menstruators are excluded from these studies? Explain the implications further.

o Do you want to add in the other side-effects the vaccine brings which have been documented widely? This allows you to point out the invisibility of menstruation/the disregard for women’s/menstruators health.

• Conclusion

o create separate section for conclusion

o List again which symptoms of menstruation the vaccine affected

Reviewer #4: (No Response)

7. PLOS authors have the option to publish the peer review history of their article (what does this mean?). If published, this will include your full peer review and any attached files.

**Do you want your identity to be public for this peer review?** For information about this choice, including consent withdrawal, please see our Privacy Policy.

Reviewer #1: **Yes: **Georgia Hales

Reviewer #4: No

---

## [Decision Letter · Decision Letter 4]

15 Mar 2024

Menstrual Changes after COVID-19 Vaccination among Menstruators of Reproductive Age: A Cross- Sectional Study from Erbil City, Iraq

PGPH-D-23-00059R4

Dear Dr Abdullah,

We are pleased to inform you that your manuscript 'Menstrual Changes after COVID-19 Vaccination among Menstruators of Reproductive Age: A Cross- Sectional Study from Erbil City, Iraq' has been provisionally accepted for publication in PLOS Global Public Health.

Best regards,

Julia Robinson

Executive Editor

Reviewer Comments (if any, and for reference):

Reviewer's Responses to Questions

**Comments to the Author**

1. If the authors have adequately addressed your comments raised in a previous round of review and you feel that this manuscript is now acceptable for publication, you may indicate that here to bypass the “Comments to the Author” section, enter your conflict of interest statement in the “Confidential to Editor” section, and submit your "Accept" recommendation.

Reviewer #1: All comments have been addressed

2. Does this manuscript meet PLOS Global Public Health’s publication criteria? Is the manuscript technically sound, and do the data support the conclusions? The manuscript must describe methodologically and ethically rigorous research with conclusions that are appropriately drawn based on the data presented.

Reviewer #1: Yes

3. Has the statistical analysis been performed appropriately and rigorously?

Reviewer #1: Yes

4. Have the authors made all data underlying the findings in their manuscript fully available (please refer to the Data Availability Statement at the start of the manuscript PDF file)?

Reviewer #1: Yes

5. Is the manuscript presented in an intelligible fashion and written in standard English?

Reviewer #1: Yes

6. Review Comments to the Author

Reviewer #1: Well done!

7. PLOS authors have the option to publish the peer review history of their article (what does this mean?). If published, this will include your full peer review and any attached files.

**Do you want your identity to be public for this peer review?** For information about this choice, including consent withdrawal, please see our Privacy Policy.

Reviewer #1: No
